# Drakenstein Child Health Study (DCHS): investigating determinants of early child development and cognition

Kirsten A Donald,[1] Michelle Hoogenhout,[1] Christopher P du Plooy,[1] Catherine J Wedderburn,[1,2] Raymond T Nhapi,[3] Whitney Barnett,[3,4] Nadia Hoffman,[5] Susan Malcolm-Smith,[6] Heather J Zar,[3,4] Dan J Stein[5,7]

► Additional material is published online only. To view please visit the journal online (http://dx.doi.org/10.1136/bmjpo-2018-000282).

## ABSTRACT

**Introduction** There is growing awareness that psychosocial risk and resilience factors in early life play a key role in influencing later health. Most work has been done in high-income settings, rather than low-income and middle-income countries (LMICs), where the majority of the global childhood population resides. The few studies with well-defined cohorts in LMICs have employed various methods and measures, making comparisons across studies challenging. This presentation describes the methodology for infant and child developmental measures used in the Drakenstein Child Health Study (DCHS), a multidisciplinary longitudinal birth cohort study in South Africa.

**Methods and analysis** We outline a multilevel approach combining a range of measures including parental reports, behaviour observations, clinician-administered scales and brain imaging. Using this approach, we aim at a longitudinal perspective of developmental, cognitive, socioemotional and neurophysiological outcomes in a birth cohort of children in an LMIC.

**Ethics and dissemination** The study was approved by the faculty of Health Sciences, Human Research Ethics Committee, University of Cape Town (401/2009), Stellenbosch University (N12/02/0002) and the Western Cape Provincial Health Research committee (2011RP45).

**Discussion** Children in the DCHS develop in a context typical of many communities in South Africa and other LMICs. There is a critical need for research in LMICs to elucidate underlying factors that inform risk for, and resilience to, poor developmental outcomes in infants born into high-risk communities. Such work may inform effective intervention strategies appropriate to this context.

## What is already known on this topic?

► There is growing awareness that psychosocial risk and resilience factors in early life play a key role in influencing later health.
► Most work has been done in high-income settings, rather than low-income and middle-income countries (LMICs), where the majority of the global childhood population resides.
► The few studies with well-defined cohorts in LMICs have employed various methods and measures, making comparisons across studies challenging.

## What this study hopes to add?

► The Drakenstein Child Health Study aims to provide an understanding of the effects of multiple risk and mitigating factors on child health and development in a LMIC.

For numbered affiliations see end of article.

**Correspondence to**
Dr Kirsten A Donald; kirsty.donald@uct.ac.za

## INTRODUCTION

Risk and resilience factors encountered during the early years of life have an enduring influence on later physiological and psychological outcomes.[1–3] A number of risk factors are already apparent in utero; for example, antepartum maternal psychological distress and depression can adversely affect infant physical, neurocognitive and socioemotional developmental outcomes.[4–6] During early childhood, exposure to stressors such as familial violence and abuse has been associated with increased risk of behaviour problems, autoimmune disorders, cardiovascular disease and premature mortality.[7–10] In low-income and middle-income countries (LMICs), key risk factors such as HIV infection and prenatal maternal malnutrition are responsible for millions of children failing to reach their full developmental potential.[11–13] Poor child outcomes may have intergenerational effects, so exacerbating their impact.[12]

At the same time, protective factors may be associated with increased resilience, and so with positive mental health and developmental outcomes in the face of stressors.[14] Resilience is thought to arise from the interplay between factors at the individual, family and community levels.[15] Protective factors can be highly context specific and can exert different effects at different time points.[15 16] Thus, longitudinal studies provide the best opportunities to identify protective factors at

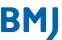

various stages of development, as well as sensitive periods for intervention. However, most studies on resilience have been done in high-income countries, where contextual factors may be different.

Indeed, the vast majority of previously reported studies have focused on psychobiological and psychosocial risk profiles in well-resourced countries. These profiles differ considerably in LMICs. For instance, there is, in general, considerably higher prevalence of low birth weight, childhood malnutrition and infectious diseases in LMICs.[17 18] In addition, critical psychosocial factors that are known to have impact on child development such as maternal depression and exposure to violence frequently have a greater prevalence in these high-risk communities.[12 13 19] There is considerable work in LMICs, including work that is longitudinal and that is culturally appropriate. However, various methods and measures have been used, so that cross-cohort comparison is not always possible.

The Drakenstein Child Health Study (DCHS) is a multidisciplinary longitudinal study investigating the early-life determinants of child health in two periurban communities in the Western Cape Province, South Africa.[20] The early-life component focuses on a broad spectrum of child health outcomes, including physical health and growth as well as neurodevelopmental, cognitive and psychological health. The study investigates the role and interaction of environmental, infectious, psychosocial, nutritional, genetic, maternal and immunological risk and protective factors for development. The DCHS follows an extensively phenotyped cohort over critical early years, aiming to provide an understanding into the effects of multiple risk and mitigating factors, and their interactions, on child health and development in a LMIC. This paper describes the methodology for the infant and child measures of the psychosocial component of the DCHS. By documenting the measures used and our reasons for choosing these, we hope to improve future harmonisation between cohort studies.

## METHODS AND ANALYSIS
### Design and setting
The DCHS is located in the periurban Drakenstein district, Western Cape, South Africa. The communities surrounding the Mbekweni and TC Newman clinics are ethnically, culturally and linguistically heterogeneous. Mbekweni consists of a mostly isiXhosa-speaking black African population, whereas TC Newman consists of a mostly Afrikaans-speaking mixed race population.[20] However, both communities are characterised by low socioeconomic status and feature a high prevalence of multiple psychosocial risk factors, including single-parent households, high rates of psychological distress and exposure to violence, HIV and substance abuse.[21] In particular, risk factors that may be highly prevalent in these communities and similar communities in the region include high rates of HIV exposure,[20 22 23] high prevalence of maternal psychological distress and depression,[5 21 24–26]

high rates of drug and alcohol usage,[21 24 27 28] high levels of violence and intimate partner violence[29] and low levels of employment and educational attainment.[21] The population is stable, with little immigration or emigration. More than 90% of people in the district use the public health system. In this regard, the communities are representative of many other communities in South Africa and other LMICs.

### Participants
Pregnant women were recruited while attending routine antenatal care at Mbekweni or TC Newman clinic between March 2012 and March 2015. Women were enrolled in the DCHS at 20–28 weeks' gestation and were followed through birth and postnatally. Pregnant mothers were eligible for the study if they were 18 years or older, planned to attend antenatal care at one of the two clinics and intended to remain in the area for at least a year. Expecting mothers provided informed written consent at enrolment and were reconsented annually following childbirth.

Mothers were provided informed consent in their preferred language, English, Afrikaans or isiXhosa, by trained study staff from the community. Informed consent forms described the scope and aims of the study, including potential harm or benefits. In total, 1137 mother–child dyads were enrolled in the study, of which four mothers had twins and one had triplets. Thus, 1143 children were enrolled in the study. A conservative cumulative attrition of 20% over 5 years was estimated, and the sample size was calculated accordingly. Enrolment criteria was broad to ensure generalisability and that the cohort would be representative of the general population. Majority of Drakenstein subdistrict births occur at Paarl Hospital, with an average of 4800 births per year; the study enrolled approximately 10% of catchment area births. Of pregnant women in the catchment area, who were provided information relating to study enrolment, 1471 mothers were determined to be ineligible (based on age, gestation or non-attendance at study clinics). A further 674 mothers were eligible but were not interested in study enrolment. Where data were available for the Cape Winelands district, the study population has comparable levels of education, partnership status and household size. Based on these sociodemographic variables and the broad enrolment criteria used, we believe that the study population is representative of the source population.

### Measures
Mothers were followed during pregnancy and childbirth. Following birth, infants and mothers returned to the clinics and were asked to complete self-report and clinician-administered measures at time points ranging from 2 weeks to 4 weeks to 5.5 years (see figure 1). At the time of submission, the oldest children in the cohort are 5 years old and the youngest children are currently 2 years old.

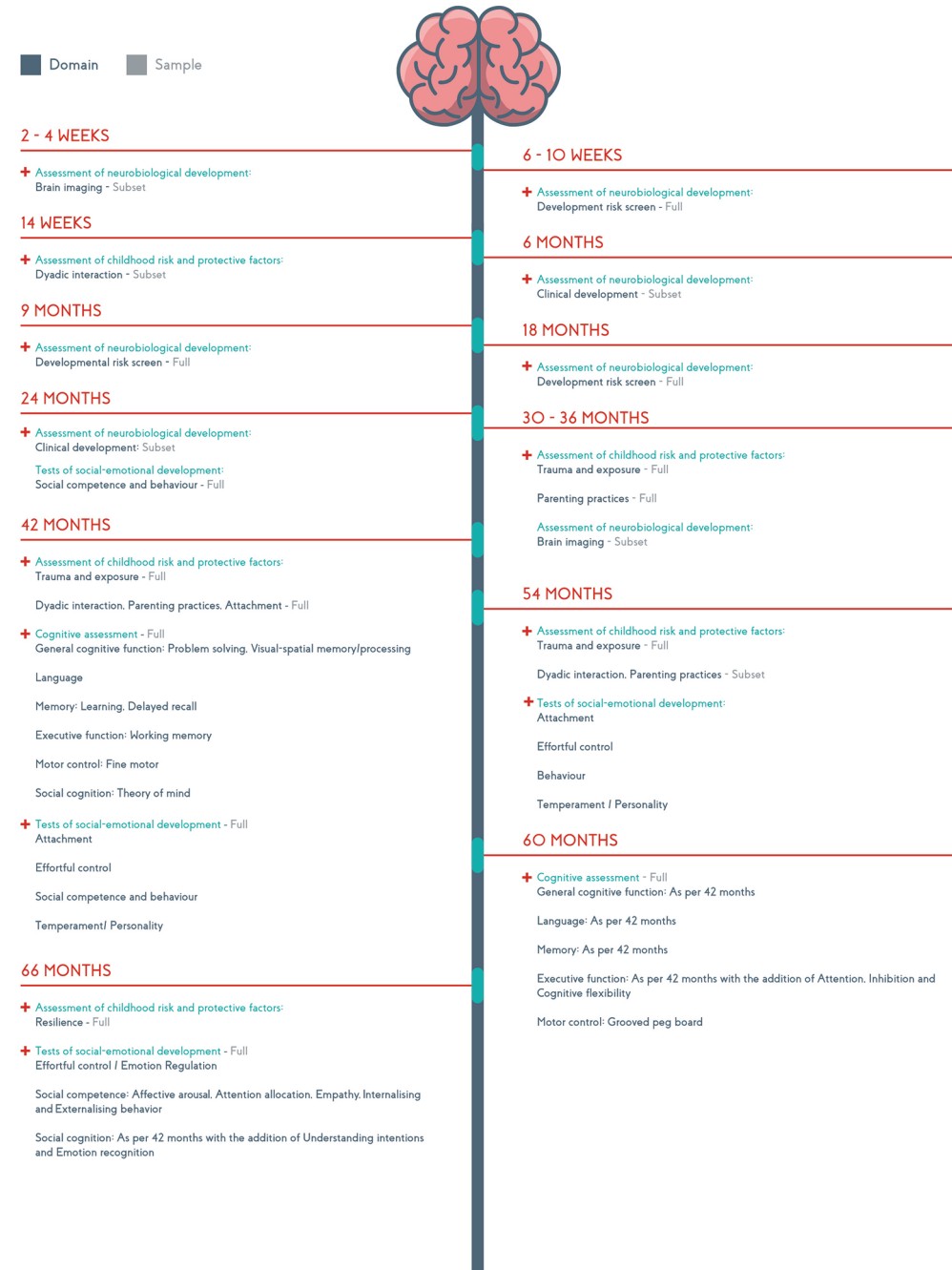

**Figure 1** Time line for child assessment.

The infant and child developmental and psychosocial measures are described here. The overview methods of the larger DCHS are described elsewhere,[20] as is the maternal and paternal psychosocial evaluation component of the study.[21] Broadly, the child measures assessed (1) social and biological risk and protective factors, (2) general neurobiologicalneurobiological (3) cognitive development and (4) socioemotional risk and protective factors for child health outcomes. Measures were translated from English to isiXhosa and using the standard forward and backwards-translation method.[30] Assessments were conducted at community centres located near the two clinics.

The tests used are detailed in table 1 with detailed description in online supplementary appendix 1. The measures of social and biological risk and protective factors (eg, exposure toviolence, alcohol and tobacco use, parenting practices, and attachment) were completed

**Table 1** Child psychosocial and neurodevelopmental measures

| Domain | Measure |
|---|---|
| Demographic data | Household income and assets, maternal education and employment. |
| Infant health information | Weight, height and diagnoses. |
| Infant neurodevelopment | MRI, DTI, MRS and fMRI. |
| AnfMRItal risk factors | Birth planning; partner support; maternal depression; alcohol, nicotine and illicit substance use; maternal childhood trauma; intimate partner violence; exposure to stressful events; psychological distress; and symptoms of peritraumatic and post-traumatic stress. |
| Trauma and exposure | |
| Emotional distress | Paediatric emotional distress scale. |
| Exposure to violence | Child exposure to community violence. |
| | Survey for exposure to community violence. |
| Parenting | |
| Dyadic interaction* | Global rating scale. |
| | Emotional availability scale. |
| Parenting practices | Parenting and family adjustment scale. |
| Attachment | Brockington postpartum bonding questionnaire. |
| Resilience | Child and youth resilience measure. |
| General development | Western Cape screen. |
| | *Bayley Scales of Infant and Toddler Development, Third Edition (Bayley-III), Third Edition* (Bayley-III). |
| General cognitive function | Kaufman Assessment Battery for Children (KABC-II). |
| Problem solving | KABC-II Conceptual thinking. |
| Visual-spatial memory/ processing | KABC-II Face recognition. |
| Visual-spatial processing and problem solving | KABC-II Triangles. |
| Working memory and motor sequencing | KABC-II Hand movements. |
| Language | Peabody Picture Vocabulary Test, Fourth Edition. |
| | KABC-II Expressive vocabulary. |
| Memory | |
| Learning | KABC-II Atlantis. |
| Delayed recall | KABC-II Atlantis delayed. |
| Executive function | |
| Working memory | Wechsler preschool and primary scale of intelligence, Fourth Edition: picture memory |
| Inhibition | Day–night task. |
| Cognitive flexibility | Dimension change card sort. |
| Attention | Test of Everyday Attention: sky search. |
| Motor control | Bayley-III: fine motor. |
| | Grooved peg board. |
| Social cognition | |
| Theory of mind | Diverse desires. |
| | Diverse beliefs. |
| | Understanding intentions. |
| | Perception-leads-to-knowledge. |
| | Location-change false belief. |
| | Unexpected contents false belief. |
| | Belief emotion. |
| | Hidden emotions. |

Continued

**Table 1** Continued

| Domain | Measure |
|---|---|
| Emotion recognition | A Developmental NEuroPSYchological Assessment (NEPSY-II): affect recognition. |
| Effortful control/emotion regulation | Snack delay. |
| | Gift-in-bag. |
| | Whisper. |
| | Rydell's emotion questionnaire. |
| Social competence | |
| Affective arousal | Pupil dilation. |
| Attention allocation | Gaze fixation. |
| Empathy | Chicago empathy for pain task. |
| | Questionnaire of cognitive and affective empathy. |
| Prosocial behaviour | Dictator game. |
| Internalising and externalising behaviour | Child behaviour checklist. |
| Temperament/personality | |
| Temperament | Rothbart Infant and Early Child Behavior Questionnaire.* |
| Callous-unemotional traits | Callous unemotional screening device. |

References for the measures are given in the appendix.
*Very short forms.
DTI, diffusion tensor imaging; fMRI, functional MRI; MRS, magnetic resonance spectroscopy.

by the mothers during several antenatal and postnatal study visits. Additionally, dyadic interactions between the mother and child were recorded. Basic demographic data and health information for the infants were obtained from participants' hospital records. The child's cognitive and general development was assessed directly at several different time points. Early child development was assessed by trained physiotherapists and occupational therapists at the clinics supervised by a paediatric neurodevelopmental specialist. Cognitive assessments, which include language, fine motor, executive functioning, memory and general and social cognitive ability, were administered by trained research assistants, with the aid of trained interpreters when necessary. Child socio-emotional development was captured through a combination of observational, self-report and parent-report measures (see online supplementary appendix 1). The socio-emotional assessment included measures of emotion regulation, affective arousal and social atten-tion-allocation, empathy, morality, prosocial behaviour, temperament and callous–unemotional traits. Additionally, a subgroup of children in the DCHS cohort underwent multimodal neuroimaging assessment at the Cape Universities Body Imaging Centre (CUBIC). The imaging modalities performed included: (1) structural MRI with T1-weighting and T2-weighting to examine cortical and subcortical volumes; (2) diffusion tensor imaging for white matter microstructure; magneticreso-nance spectroscopy; and (4) resting state functional MRI for regional connectivity.

### Descriptive statistics
We use descriptive statistics (medians, IQRs, counts and percentages) to present the sociodemographicstatis-tics on sites TC Newman and Mbekweni. $\chi^2$ and Mann-Whitney tests were used to test for differences between the sites. As the study is still ongoing, the data presented here was collected antenatally and at birth.

### Child sociodemographics
The cohort followed 1137 mothers over the course of 3 years during the initial recruitment period. During this time, there was a total of 1143 live births including four sets of twins and one set of triplets (see table 2). Within this group, 17.2% of children were born preterm (definepretermas less than 37 weeks' gestation). The birth weight and lengths of the children have been converted to z-scores according to WHO standardisation and adjusted for gestation. In line with our previous reports,[31 32] the children at TC Newman clinic were born significantly smaller in weight compared with Mbekweni children; however, the IQR falls within two z-scores of 0 in both clinics.

### Maternal and family sociodemographics
The maternal and family characteristics are presented in table 3. Socioeconomic status was low across both clinics as shown by the low household income levels, maternal employment rates of 27% and maternal educational attainment. The median maternal age at enrolment was 26 years (IQR 22–31), and the majority of mothers had completed some secondary schooling by this point. The mothers reported that over 65% of

**Table 2** Child sociodemographisociodemographics

| Variable | Mbekweni | TC Newman | Total | P values |
|---|---|---|---|---|
| | **Number (% of each clinic)** | | | |
| Mothers | 628 | 509 | 1137 | |
| Live births | 634 | 509 | 1143 | |
| Twin sets | 4 | 0 | 4 | |
| Triplet sets | 1 | 0 | 1 | |
| Preterm births | 107 (16.88) | 83 (16.31) | 190 (16.32) | 0.797 |
| Child's race | | | | |
| Black | 627 (99.05) | 6 (0.95) | 633 (55.38) | <0.001 |
| Mixed | 7 (1.11) | 503 (98.82) | 510 (44.62) | |
| | **Median (IQR)** | | | |
| Birth weight z-score* | −0.4 (−1.2–0.2) | −0.7 (−1.4 to −0.1) | −0.5 (−1.3–0.1) | <0.001 |
| Birth length z-score* | 0.1 (−0.9–1.0) | −0.03 (−0.9–0.8) | 0.003 (−0.9–0.9) | 0.151 |

*Adjusted for gestation.

**Table 3** Maternal and family sociodemographisociodemographics

| Variable | Cape Winelands* n=787 490 | Mbekweni | TC Newman | Total | P values |
|---|---|---|---|---|---|
| | | **Number (% of each clinic)** | | | |
| Maternal educational attainment | | | | | 0.038 |
| Primary | | 49 (7.8) | 37 (7.3) | 86 (7.6) | |
| Some secondary | | 340 (54.1) | 266 (52.3) | 606 (53.3) | |
| Completed secondary | 193 723 (24.6) | 189 (30.1) | 183 (36.0) | 372 (32.7) | |
| Any tertiary | 82 686 (10.5) | 50 (8.0) | 23 (4.5) | 73 (6.4) | |
| Currently employed | 675 666 (85.8) | 157 (25.0) | 149 (29.3) | 306 (27.0) | 0.106 |
| Married/cohabiting | 282 709 (35.9) | 237 (37.8) | 221 (43.4) | 458 (40.3) | 0.054 |
| Unplanned pregnancy | | 366 (68.0) | 286 (62.7) | 652 (65.6) | 0.079 |
| Partner support | | | | | <0.001 |
| Not at all/slightly supportive | | 42 (7.8) | 46 (10.13) | 88 (8.9) | |
| Moderately supportive | | 86 (16.0) | 19 (4.2) | 105 (10.6) | |
| Considerably/extremely supportive | | 409 (76.1) | 389 (85.7) | 798 (80.5) | |
| Reliance on partner for help | | | | | <0.001 |
| Not at all/not very often | | 53 (9.9) | 46 (10.1) | 99 (10.0) | |
| Some of the time | | 170 (31.7) | 28 (6.2) | 198 (20.0) | |
| Most/all of the time | | 314 (58.5) | 380 (83.7) | 694 (70.0) | |
| Monthly income | 119 536/year (average) | | | | <0.001 |
| <R1000 (<$75) | | 263 (41.9) | 167 (32.8) | 430 (37.8) | |
| R1000–5000 ($75–374) | | 299 (47.6) | 254 (49.9) | 553 (48.6) | |
| >R5000 (>$374) | | 66 (10.5) | 88 (17.3) | 152 (13.5) | |
| | | **Median (IQR)** | | | |
| Household size | 3.7 persons | 4 (3–6) | 5 (4–7) | 4 (3–6) | <0.001 |
| Mother's age at enrolment | | 27 (22–32) | 25 (21–29) | 26 (22–31) | <0.001 |

Census 2011 Municipal Report Western Cape. Statistics South Africa. Report no. 03-01-49. http://www.statssa.gov.za.

pregnancies were unplanned. Approximately 40% were currently married or cohabiting with their partner, and a high proportion reported that partners were supportive, although this differed across clinics.

Mothers and children in both communities were frequently exposed to community violence (see table 4). Levels of violence exposure were higher in TC Newman than in Mbekweni, but both communities from this

**Table 4** Family risk and protective factors

| Variable | Mbekweni<br>Mean (SD) | TC Newman | Total | P values |
|---|---|---|---|---|
| PAFAS Parenting | | | | |
| Consistency | 7.95 (2.01) | 6.24 (2.92) | 7.17 (2.61) | <0.001 |
| Coercive parenting | 6.24 (3.38) | 5.19 (3.98) | 5.76 (3.70) | <0.001 |
| Encouragement | 1.23 (1.66) | 1.00 (1.50) | 1.13 (1.59) | 0.113 |
| Parent–child relationship | 0.63 (1.48) | 1.37 (1.76) | 0.97 (1.65) | <0.001 |
| PAFAS Family Adjustment | | | | |
| Parental adjustment | 2.72 (2.67) | 2.36 (2.86) | 2.56 (2.76) | 0.025 |
| Family relationships | 1.46 (2.01) | 2.45 (2.85) | 1.91 (2.48) | <0.001 |
| Parental teamwork | 1.32 (1.67) | 1.93 (2.08) | 1.59 (1.89) | <0.001 |
| SECV Total | 21.27 (6.57) | 26.92 (7.46) | 23.84 (7.53) | <0.001 |
| CECV Total | 38.65 (3.88) | 40.13 (4.21) | 38.65 (3.88) | <0.001 |

*Note:* the questionnaires were administered at 2.5 years. Higher scores on the PAFAS indicate higher levels of dysfunction, that is, higher consistency scores indicate less consistent parenting and higher coercive parenting scores indicate more coercive parenting.
CECV, Child Exposure to Community Violence Checklist; PAFAS, Parenting and Family Adjustment Scale; SECV, Survey for Exposure to Community Violence.

cohort reported greater exposure to violence than reported in a previous study in South Africa.[33] Exposure to violence is thus highly prevalent in these communities both within the home and external community environment. Both communities reported less consistent parenting (ie, higher consietency scores) but otherwise similar parenting styles and family adjustment, compared with European samples.[34] Mothers in Mbekweni reported more coercive parenting, less consistency and less encouragement, but also better family relationships and parental teamwork than in TC Newman.

## DISCUSSION

This paper highlights the rationale and approach to assessing both psychosocial risk and protective factors impacting the development of children in a high-risk South African communities. The study follows a multilevel approach that targets developmental, cognitive, socioemotional socioemotionalcal outcomes. As can be seen from some of our baseline data ($n = 1143$), there are a broad number of sociodemographic risk and resilience factors for children in this region. Demographic and sociodemographic data show that although these communities are in close proximity, they differ substantively in social and financial resources.

Given these sorts of risk and resilience factors, it is important to assess child outcomes using a multidimensional approach.[35] This includes three important components that are built into this cohort. First, the DCHS collects biomedical and psychosocial risk factors across a wide range of factors in the prenatal period and first years of life. These include both factors that are known to put children at risk for poor outcome such as maternal mental health, substance use disorders, poor nutrition and factors known to be protective or hypothesised to

potentially support development including early infant feeding choices, pregnancy support and maternal bonding and attachment styles. Second, the outcome measures described in this manuscript are also multidimensional and allow examination of outcomes in terms of the dyadic relationship and the family system into which these children are born. Third, the cohort is following these mother infant pairs over time. Longitudinal data (with repeat measures) allows the investigation of developmental, cognitive and socioemotional trajectories as well as the interactions between exposures within the context of this cohort. The investigation of the timing of maximum windows of vulnerability becomes possible with this approach.

Attention to the ethical issues requires consideration in a study of this type. The DCHS maintains an active programme focused on community engagement, including regular engagement with study participants for feedback on study involvement, active dissemination of research results to key local stakeholders and distribution of health promotion information to study participants. Given the context, a key ethical obligation includes screening within the study population for physical and mental health issues, in both mothers and children. Screening is done in the DCHS in conjunction with an active referral system and is bolstered by close relationships between study staff and provincial health staff. All women involved in the study, independent of identified mental or physical health problems, receive information regarding service providers in the area. The network of investigators in the DCHS with strong and relevant clinical background in the South African public health environment is a strength of the research and has allowed realistic and integrated referral systems to be developed and implemented as part of the study.

Very few cohorts are reported that take into account the composite effects of multiple factors on health and development in the early years. This is especially true of cohorts from LMICs where young children are exposed to overlapping epidemics of infectious and non-communicable diseases. The P-MaMie birth cohort in Ethiopia[36] collected information on maternal mental health as well as growth and developmental outcomes in very young infants but represents an almost entirely rural community in Africa with attendant risk factors likely to vary from the periurban community described in the DCHS. With urbanisation representing a critical current epidemiological phenomenon, documenting communities in this context becomes increasingly important. The Pelotas cohorts in Brazil[37] represent one of the longest standing set of population-based cohorts in the global south. These are large cohorts capturing whole communities have been documenting maternal and child outcomes for over 20 years. The most recent of these cohorts starting in 2015 is the first of these to collect prospective antenatal data on the mothers. Though smaller, the DCHS, at present, has been able to collect the most comprehensive set of biological, psychosocial, environmental, maternal and child data and so carefully measures outcomes through use of sensitive modalities including brain imaging in a high-risk setting. Sensitivity to exposures, individually and together, in both external and internal environments, are different at specific ages and periods in development. The developmental window spanning the 40 weeks of pregnancy and the first years of life appear to be a critical period where environmental exposure may cause embedded effects that may have impact across the lifespan. There is a critical need for research in this field to elucidate the underlying factors that inform risk for and resilience from poor developmental outcomes in infants born into high risk communities that may ultimately inform effective preventative and ameliorating intervention strategies appropriate to this context. From a public health perspective, a better understanding of the relevant mechanisms is critical, as this may ultimately drive preventative and targeted therapeutic approaches.

The United Nation's Sustainable Development Goals (SDG) were officially adopted in 2015. These cross-cutting SDGs consistently forefront the importance of programmes targeting maternal and child health (in particular, the theme of early child growth and development), as being key in the global strategy to optimise human health and well-being across the lifespan. In the South African setting, where children make up around a third of the population, expectant mothers and their young infants are a particularly important focus. Much more attention is needed to address maternal and infant health, in order to decrease early mortality and later morbidity in this vulnerable population.

## LIMITATIONS

Limitations of the study include the fact that though extremely well characterised, the cohort is a modest size. Though measures investigating aspects of child health, development and cognition have been administered to as much of the cohort as possible, certain measures have been administered only on a subset of the group (eg, neuroimagineg dneuroimagingipant burden and the cost of assessment. Although care was taken to translate measures into Afrikaans and isiXhosa, there willalways be some difficulty in interpreting the results of measures designed in a different language and cultural context. Every effort was made to use tools that minimised problems in this area.

**Author affiliations**
[1]Division of Developmental Paediatrics, Department of Paediatrics and Child Health, Red Cross War Memorial Children's Hospital, University of Cape Town, Cape Town, South Africa
[2]Department of Clinical Research, London School of Hygiene & Tropical Medicine, London, UK
[3]Unit on Child and Adolescent Health, South African Medical Research Council, Cape Town, South Africa
[4]Department of Paediatrics and Child Health, Red Cross War Memorial Children's Hospital, University of Cape Town, Cape Town, South Africa
[5]Department of Psychiatry and Mental Health, University of Cape Town, Cape Town, South Africa
[6]Applied Cognitive Science and Experimental Neuropsychology Team, Department of Psychology, University of Cape Town, Cape Town, South Africa
[7]Unit on Risk and Resilience in Mental Disorders, South African Medical Research Council (SAMRC), Cape Town, South Africa

**Acknowledgements** We would like to thank the study staff in Paarl, the study data team and lab teams, the clinical and administrative staff of the Western Cape Government Health Department at Paarl Hospital and at the clinics for support of the study. We acknowledge the advice from members of the study International Advisory Board and our collaborators. We would like to thank the families and children who participated in this study.

**Contributors** This study requires multidisciplinary expertise in the execution of measures of this type. DJS is PI of the psychosocial aspects of the Drakenstein Child Health Study (DCHS) cohort and contributed to the design and decision making involving psychosocial tools and measures used as well as general study design. HJZ is PI of the umbrella DCHS cohort and played a central role in the operational aspects and design of the study. KAD, PI of the child psychosocial aspects of the study, was involved in the design of the study and operational aspects of the study and played a key role in the child psychosocial measures used. SM-S, MH and CPdP contributed to decision making involving tools used, training and operational aspects of the child assessments. NH, WB and CJW contributed to the operational aspects of the study, QC of data described and data management. RTN contributed to data management. Authors contributed to sections relating to their area of expertise in the manuscript. All authors reviewed and approved the final version of this manuscript.

**Funding** This study was funded by National Institute on Alcohol Abuse and Alcoholism (R21AA023887), US Brain and Behaviour Foundation (24467) and Bill and Melinda Gates Foundation (OPP 1017641), Collaborative Initiative on Fetal Alcohol Spectrum Disorders (U24 AA014811), Medical Research Council of South Africa, South African Medical Research Council, Wellcome Trust (203525/Z/16/Z), National Research Foundation, and Newton Fund (NAF002\1001).

**Competing interests** None declared.

**Patient consent** Parental/guardian consent obtained.

**Ethics approval** The study was approved by the faculty of Health Sciences, Human Research Ethics Committee, University of Cape Town (401/2009), Stellenbosch University (N12/02/0002) and the Western Cape Provincial Health Research committee (2011RP45).

**Provenance and peer review** Not commissioned; externally peer reviewed.

**Data sharing statement**  Collaborations for the analysis of data are welcome. The DCHS has a large and active group of investigators and postgraduate students, and many have successfully partnered with researchers from other institutions. In particular, we encourage collaborations that lead to skills transfer and capacity building for postgraduate students. Researchers who are interested in datasets or collaborations can contact the PI, Heather Zar (heather.zar@uct.ac.za) with a concept note outlining the request. More information can be found on our website (http://www.paediatrics.uct.ac.za/scah/dclhs).

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
