## [Reviewer comments · BMJ Paediatrics Open]

ARTICLE DETAILS

TITLE (PROVISIONAL)	Protocol: The Drakenstein Child Health Study (DCHS)-Investigating determinants of early child development and cognition
AUTHORS	Donald, Kirsten; Hoogenhout, Michelle; du Plooy, Christopher; Wedderburn, Catherine; Nhapi, Raymond; Barnett, Whitney; Hoffman, Nadia; Malcolm-Smith, Susan; Zar, Heather; Stein, Dan

VERSION 1 – REVIEW

REVIEWER	Reviewer name Professor Raghu Lingam Institution and Country UNSW Australia Competing interests None
REVIEW RETURNED	15-Mar-2018

GENERAL COMMENTS	This is an important study looking at multiple risk factors on a child's development in a low and middle income country context. The paper will act as a protocol paper for subsequent analyses that the team are wanting to perform. The study is an antenatal recruited longitudinal cohort study. What is unclear is how representative this cohort is compared to this area of South Africa, and South Africa as a whole. The number of women recruited into the study is currently not compared to the total number of births in this area from 2012-2015 (period of recruitment). This comparison is essential for us to understand this cohort compared to the entire local population. I strongly recommend that these figures are inserted and Table 3 amended to compare the cohort to the general population during this time. If data is available comparing the cohort to those that refused to enrol this would also be helpful though not essential. How measures were administered needs to be outlined in the main body of the text. Discussion needs to place this cohort study in the context of other LAMIC cohort studies and specifically outline what is new and unique about this study.
--

VERSION 1 – AUTHOR RESPONSE

Reviewer: 1
Reviewer name: Professor Raghu Lingam

Reviewer: 1
Institution and Country: UNSW Australia

Reviewer: 1

Comment: This is an important study looking at multiple risk factors on a child's development in a low and middle income country context. The paper will act as a protocol paper for subsequent analyses that the team are wanting to perform.

The study is an antenatal recruited longitudinal cohort study. What is unclear is how representative this cohort is compared to this area of South Africa, and South Africa as a whole. The number of women recruited into the study is currently not compared to the total number of births in this area from 2012-2015 (period of recruitment). This comparison is essential for us to understand this cohort compared to the entire local population. I strongly recommend that these figures are inserted and Table 3 amended to compare the cohort to the general population during this time. If data is available comparing the cohort to those that refused to enrol this would also be helpful though not essential.

Response: We thank Professor Lingam for his review and for highlighting these important points. We have sourced census data from the region and added to table 3. We have added the following contextualizing text to the manuscript.

Enrolment criteria was broad to ensure generalizability and that the cohort would be representative of the general population. Majority of Drakenstein sub-district births occur at Paarl Hospital, with an average of 4800 births per year, the study enrolled approximately 10% of catchment area births. Of pregnant women in the catchment area, who were provided information relating to study enrolment, 1471 mothers were determined to be ineligible (based on age, gestation or non-attendance at study clinics). A further 674 mothers were eligible but were not interested in study enrolment. Where data available for the Cape Winelands district, the study population has comparable levels of education, partnership status and household size. Based on these sociodemographic variables and the broad enrolment criteria used, we believe that the study population is representative of the source population."

Comment: How measures were administered needs to be outlined in the main body of the text.

Response: Additional text has been added to the main body of the manuscript that expand on the measure description. The added text is copied below for ease of reference. It is also highlighted in a different colour in the main body of the text. We have also made a small correction to one of the assessment time points and amended the manuscript and figure accordingly.

"The tests used are detailed in Table 1 with detailed description in supplementary appendix. The measures of social and biological risk and protective factors (e.g. exposure to violence, alcohol and tobacco use, parenting practices, and attachment) were completed by the mothers during several antenatal and postnatal study visits. Additionally, dyadic interactions between the mother and child were recorded. Basic demographic data and health information for the infants were obtained from participants' hospital records. The child's cognitive and general development was assessed directly at several different time points. Early child development was assessed by trained physio and occupational therapists at the clinics supervised by a paediatric neurodevelopment specialist. Cognitive assessments which include language, fine motor, executive functioning, memory and general and social cognitive ability, were administered by trained research assistants, with the aid of trained interpreters when necessary. Child socio-emotional development was captured through a combination of observational, self-report and parent-report measures (see supplementary materials). The socio-emotional assessment included measures of emotion regulation, affective arousal and social attention-allocation, empathy, morality, prosocial behaviour, temperament and callous-unemotional traits. Additionally, a subgroup of children in the Drakenstein Child Health Study cohort underwent multimodal neuroimaging assessment at the Cape Universities Brain Imaging Centre (CUBIC). The imaging modalities performed included: (1) structural magnetic resonance imaging (MRI) with T1 and T2-weighting to examine cortical and subcortical volumes;

(2) diffusion tensor imaging (DTI) for white matter microstructure; (3) magnetic resonance spectroscopy (MRS) and; (4) resting state functional MRI for regional connectivity.”

Comment: Discussion needs to place this cohort study in the context of other LAMIC cohort studies and specifically outline what is new and unique about this study.

Response: Thank you for this recommendation. Amendments have been made pulling in some contextualizing discussion around other cohorts.

“The P-MaMie birth cohort in Ethiopia³⁶ collected information on maternal mental health as well as growth and developmental outcomes in very young infants, but represent an almost entirely rural community in Africa with attendant risk factors likely to vary from a peri-urban community described in the DCHS. With urbanisation representing a critical current epidemiological phenomenon, documenting communities in this context becomes increasingly important. The Pelotas cohorts in Brazil³⁷ represent one of the longest standing set of population based cohorts in the global South. These are large cohorts capturing whole communities have been documenting maternal and child outcomes for over 20 years. The most recent of these cohorts starting in 2015 is the first of these to collect prospective antenatal data on the mothers. Though smaller, the DCHS, at present has been able to collect the most comprehensive set of biological, psychosocial, environmental, maternal and child data and so carefully measures outcomes through use of sensitive modalities including brain imaging in a high-risk setting.”